# LEARN ROBUST FEATURES VIA ORTHOGONAL MULTI-PATH

## ABSTRACT

It is now widely known that by adversarial attacks, clean images with invisible perturbations can fool deep neural networks. To defend adversarial attacks, we design a block containing multiple paths to learn robust features and the parameters of these paths are required to be orthogonal with each other. The so-called Orthogonal Multi-Path (OMP) block could be posed in any layer of a neural network. Via forward learning and backward correction, one OMP block makes the neural networks learn features that are appropriate for all the paths and hence are expected to be robust. With careful design and thorough experiments on e.g., the positions of imposing orthogonality constraint, and the trade-off between the variety and accuracy, the robustness of the neural networks is significantly improved. For example, under white-box PGD attack with $l_\infty$ bound $8/255$ (this is a fierce attack that can make the accuracy of many vanilla neural networks drop to nearly $10\%$ on CIFAR10), VGG16 with the proposed OMP block could keep over $50\%$ accuracy. For black-box attacks, neural networks equipped with an OMP block have accuracy over $80\%$. The performance under both white-box and black-box attacks is much better than the existing state-of-the-art adversarial defenders.

## 1 INTRODUCTION

In recent years, Deep Neural Networks (DNNs) have been widely applied in many fields (Goodfellow et al., 2016). Despite the great progress, vulnerability of DNNs has also been found. For example, in classification task in computer vision, by adding well-designed, visually-imperceptible perturbations on clean images, the resulting perturbed images, a.k.a. *adversarial examples*, can successfully fool many well-trained DNNs (Szegedy et al., 2014; Goodfellow et al., 2015). Such a process of generating adversarial examples is called *adversarial attack*.

Since its proposal, there have been many interesting adversarial attacks, which can be categorized into two types, *black-box attack* (Papernot et al., 2017; Liu et al., 2017; Chen et al., 2017; Su et al., 2019) and *white-box attack* (Goodfellow et al., 2015; Kurakin et al., 2017; Carlini & Wagner, 2017; Madry et al., 2018; Tang et al., 2019). As the name suggests, white-box attacks need complete information of the target model. While black-box attacks rely on the output of target model or transferability across models.

For a neural network $f(\mathbf{x}; \theta)$ with input $\mathbf{x}$ and parameters $\theta$, we denote the trained parameters as $\hat{\theta}$ and the example to be attacked as $\mathbf{x}_0$. Adversarial attack tries to find a small $\Delta\mathbf{x}$ such that $f(\mathbf{x}_0 + \Delta\mathbf{x}; \hat{\theta}) \neq f(\mathbf{x}_0; \hat{\theta})$. To defend the attack, i.e., to keep the both sides equal, adversarial training (Szegedy et al., 2014; Goodfellow et al., 2015; Madry et al., 2018) includes a group of adversarial perturbations in training process to keep $f(\mathbf{x}_0 + \Delta\mathbf{x}; \theta) = f(\mathbf{x}_0; \theta)$. Generally speaking, adversarial training is the most efficient defence strategy until now (Athalye et al., 2018; Tramer et al., 2020), but the attack needs to be known in advance. To adapt to all perturbations, researchers consider the response to perturbations $\Delta\mathbf{x}$ in $\mathcal{B}_\varepsilon = \{\Delta\mathbf{x} | \|\Delta\mathbf{x}\| \leq \varepsilon\}$. If the maximum change is small, then a certified robustness could be guaranteed (Raghunathan et al., 2018; Cohen et al., 2019).

Motivated by the coupling of samples and parameters, one could also impose randomness on parameters to enhance the robustness. Consider a linear layer, which includes convolution layer and fully-connected layer. Imposing perturbation into samples is equal to giving randomness to parameters: $\forall \Delta\mathbf{x} \in \mathcal{B}_\varepsilon$, there exists $\Delta\theta$ that satisfies $\langle \mathbf{x}_0 + \Delta\mathbf{x}, \hat{\theta} \rangle = \langle \mathbf{x}_0, \hat{\theta} + \Delta\theta \rangle$. Pioneering and

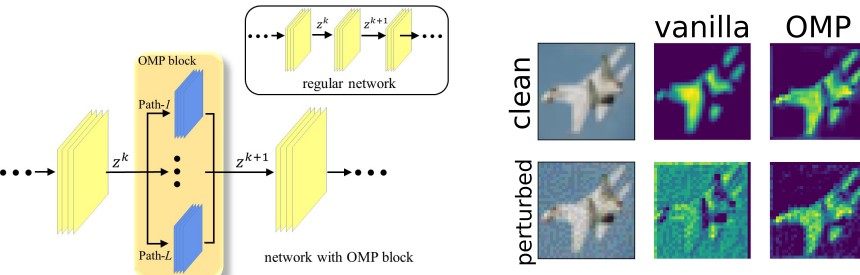

(a) network with OMP block v.s. regular network  (b) learned features (1st layer, VGG16)

Figure 1: (a) OMP block is to replace a single path by multiple ones and can be posed anywhere in a neural network. (b) When clean examples (1st row) are adversarially perturbed (2nd row, generated by attacking a third neural network), the features learned by a vanilla VGG16 change a lot. After imposing an OMP block even in the last layer, the learned features by the first layer become much more robust.

representative works on this direction could be found in (He et al., 2019; Liu et al., 2018b), which aim to learn a distribution for the parameters. The advantages include network diversity and low dependence on the attacks. However, its learning is not very effective: the learned distribution tends to shrink to one optimal solution.

In this paper, we propose to embed multiple paths into a neural network. In a regular neural network, a block with input $\mathbf{z}^k$ and output $\mathbf{z}^{k+1}$ is denoted as $\mathbf{z}^{k+1} = g(\mathbf{z}^k; \theta_g)$. For the mapping $g$, we will train multiple paths $g^i(\mathbf{z}^k; \theta_{g^i})$, then it could give multiple outputs and the rest layers are trained to adapt all the paths. A key issue here is that we require the parameters of all paths orthogonal to each other, which guarantees the diversity and coverage. Fig.1(a) gives a comparison illustration of a regular network and a network embedded with $L$ paths.

The proposed Orthogonal Multi-Path (OMP) block can be posed in any layer of a neural network. It is not surprising that the *follow-up layers* are more robust since they are capable to handle features from multiple paths. Let us consider a simple example. A VGG16 is trained on CIFAR10 and we put an OMP block in the first layer. The average feature change $\|\Delta \mathbf{z}^{k+2}\|_\infty$ at the layer after OMP block caused by perturbations $\|\Delta \mathbf{x}\|_\infty \leq 8/255$ on test images is bounded by 1.39 (which is actually a certificated robustness measure) in the vanilla VGG16 and is improved to 0.82 by the OMP block. Interestingly, OMP is also helpful for the *front layers*. In Fig.1(b), we visualize the learned features after the first layer in a vanilla network and the same network with an OMP block posed on the last layer. Although the OMP block is posed on the final layer, it could *correct* the learned features at the first layer, resulting in much more alike features of clean and perturbed images. This phenomenon is explained by the **backward correction** theory proposed by Allen-Zhu & Li (2020). Actually, this result could be a strong evidence to verify that training higher-level layers improves the features of lower-level ones.

With forward learning and backward correction, OMP block could make the feature extractor adaptive to multiple paths and then enhance the robustness of the whole networks. For example, under white-box PGD attack with $l_\infty$ bound $8/255$, which could destroy the accuracy of many vanilla networks to nearly $10\%$ on CIFAR10, VGG16 with the proposed OMP block could keep over $50\%$ accuracy. The contributions of this work are summarized as follows:

- A novel defence method is proposed, which introduces orthogonal multiple paths into a neural network to enhance the robustness.

- Extensive empirical results against different white-box and black-box attacks indicate the superior robustness of networks with OMP block in vanilla and adversarial training.

- A thorough empirical analysis on different positions of the OMP block is provided, illustrating the distinct properties. Ablation study also demonstrates the necessity and effectiveness of the OMP block.

## 2 RELATED WORK

### 2.1 ADVERSARIAL ATTACK

White-box adversarial attacks are generally based on the gradient information. Given a DNN $f(\cdot)$ and loss function $\mathcal{L}(\cdot, \cdot)$, Goodfellow et al. (2015) propose a gradient-based white-box attack method, named Fast Gradient Sign Method (FGSM), which generates an adversarial example $\hat{\mathbf{x}}$ from a clean sample $\mathbf{x}$ with the true label $y$ as follows,

$$\hat{\mathbf{x}} = \mathbf{x} + \varepsilon \mathrm{sign}(\nabla_{\mathbf{x}} \mathcal{L}(f(\mathbf{x}), y)) \tag{1}$$

where $\varepsilon$ is the perturbation step size. Kurakin et al. (2017) then propose two iterative variants of FGSM, basic iterative method and iterative least-likely class method. Projected Gradient Descent (PGD) attack proposed by Madry et al. (2018) is the strongest attack until now. Starting from $\hat{\mathbf{x}}^{(0)} = \mathbf{x}$, the iterative generation process of PGD at the $(t+1)$-th iteration can be described as,

$$\hat{\mathbf{x}}^{(t+1)} = P_{\gamma} \Big\{ \hat{\mathbf{x}}^{(t)} + \varepsilon \mathrm{sign}(\nabla_{\mathbf{x}} \mathcal{L}(f(\hat{\mathbf{x}}^{(t)}), y)) \Big\} \tag{2}$$

where $P_{\gamma}$ is the projection to the set $\{\mathbf{x} | \|\mathbf{x} - \hat{\mathbf{x}}^{(0)}\|_{\infty} \leq \gamma\}$ and $\varepsilon$ is the step size in each update.

Black-box adversarial attacks only have access to the input and output of a model. To conduct black-box attacks, a typical method based on the transferability of adversarial examples is to first attack a substitute model, which is transparent to the attackers. Then, the adversarial examples from the substitute model are used to attack the target model (Liu et al., 2017; Papernot et al., 2017). Apart from this, other black-box attacks aim at estimating the gradients by keep querying the target network (Chen et al., 2017; Su et al., 2019).

### 2.2 ADVERSARIAL DEFENCE

To improve the adversarial robustness of DNN models, training based on adversarial examples, called adversarial training, is efficient. Among different adversarial examples generation algorithms, PGD-based adversarial training has been proved the best choice and keeps the highest accuracies under different attacks (Madry et al., 2018; Athalye et al., 2018). Other defence methods include network distillation (Papernot et al., 2016), adversarial detection (Lu et al., 2017; Metzen et al., 2017) and feature analysis (Xu et al., 2018; Xie et al., 2019), which are suitable to different situations.

Another line of defences is to introduce randomness on parameters motivated by the coupling of samples and parameters. For example, He et al. (2019) propose to add learnable Gaussian noise on parameters, and Liu et al. (2018b) combine adversarial training with Bayes neural network. Apart from learning a distribution for the parameters, researchers also try to inject randomness into neural networks in other ways: such as fusing the activations with fixed Gaussian noise (Liu et al., 2018a) or learnable Gaussian noise (Jeddi et al., 2020), performing dropout with a weighted distribution (Dhillon et al., 2018), and randomizing the input images (Xie et al., 2018). However, Athalye et al. (2018) claim that the improvements brought by these methods are due to stochastic gradients.

Last but not least, we notice that very recent concurrent work (Jalwana et al., 2020) also uses orthogonality to enhance the robustness of neural networks. The proposed OMP block is totally different from (Jalwana et al., 2020): 1) Jalwana et al. (2020) train a new regular network that is orthogonal with a reference network at every layer. While we pose an OMP block in any layer of a network, and the orthogonality is imposed on the inner paths in OMP block; 2) Jalwana et al. (2020) require orthogonality on the gradients, while OMP block imposes orthogonality on the parameters; 3) Jalwana et al. (2020) only investigate the resistance against adversarial examples created from the reference network, while networks with OMP block could defend both white-box and black-box attacks.

## 3 METHODOLOGY

### 3.1 ORTHOGONAL MULTI-PATH

We embed an OMP block into a neural network. This OMP block could be posed at any layer and contains multiple mutually-orthogonal paths. Not only the learned features in the follow-up layers show better robustness, but also the OMP block could correct features learned by front layers.

Without loss of generality and to make the notation simpler, we take the OMP block posed on the last linear layer as an example to derive the objective function, where each path in the OMP block contains a linear classifier (see Fig.2(c)). In a regular network, the linear classifier is denoted as $g(\cdot)$ with weight matrix $\mathbf{W}$ (assuming no bias term), and the remainder of the network is denoted as $h(\cdot)$. Thus the total network is denoted as $g(h(\cdot)) : \mathcal{R}^d \to \mathcal{R}^K$, where $d$ and $K$ are the input and output dimension respectively.

Before stepping into the objective function, we first give the definition of orthogonality. Generally, two one-dimensional vectors are orthogonal if their inner product equals to zero. For two matrices or tensors, we first vectorize them and then perform ordinary inner product. The inner product of two linear classifiers $g^i$ and $g^j$ is computed as:

$$\langle g^i, g^j \rangle \equiv \langle \text{vec}(\mathbf{W}^i), \text{vec}(\mathbf{W}^j) \rangle. \tag{3}$$

Two linear classifiers are orthogonal if the inner product of their vectorized weight matrices equals to zero.

To train a neural network with an OMP block posed on the last linear layer, we have the following optimization problem:

$$\min_{h, g^1, ..., g^L} \sum_{i=1}^{L} \mathcal{L}(g^i(h(\mathbf{x})), y) \tag{4}$$
$$\text{s.t.} \quad \langle g^i, g^j \rangle = 0, \quad \forall i \neq j, \quad i, j \in \{1, 2, ..., L\}$$

where $L$ denotes the number of paths in the OMP block. The objective function in Eq.(4) indicates that we minimize the sum of classification losses of all the networks w.r.t. every path in the OMP block. The equality constraints in Eq.(4) require that these classifiers on the paths in the OMP block have to be mutually-orthogonal. Parameters to be optimized include all the paths in the OMP block and the remainder of the network. Incorporating the equality constraints into the objective function, we have the following optimization problem,

$$\min_{h, g^1, ..., g^L} \sum_{i=1}^{L} \mathcal{L}(g^i(h(\mathbf{x})), y) + \lambda \sum_{i=1}^{L} \sum_{j=1, j \neq i}^{L} \langle g^i, g^j \rangle^2, \tag{5}$$

where $\lambda$ is the orthogonality coefficient. In Eq.(5), the inner product term gets squared and summed to achieve the orthogonality of every two paths. For cases where OMP block is imposed on other layers (see Fig.2(a) and Fig.2(b)), the optimization problem has a similar form as Eq.(4) and Eq.(5) and is omitted.

## 3.2 TRAINING AND INFERENCE

Alg.1 illustrates an example to train a neural network with one OMP block posed on the last linear layer. The orthogonality constraint can be seamlessly equipped with vanilla training or adversarial training. In vanilla training, at each iteration, we compute the loss values of Eq.(5), back propagate the gradients and update parameters. In adversarial training, at each iteration, after updating parameters using clean samples, adversarial examples are generated based on current network, and then

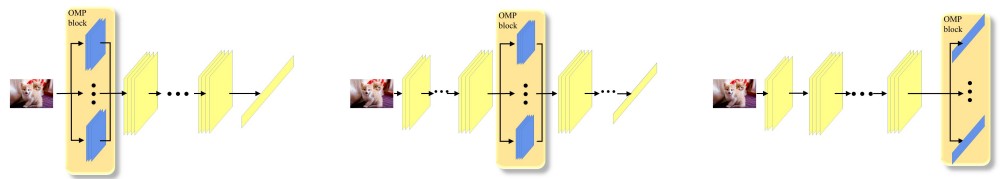

(a) OMP block on the first convolu-  (b) OMP block on the middle con-  (c) OMP block on the last linear
tion layer.                          volution layer.                     layer.

Figure 2: Illustration of three architectures of networks with one OMP block. The parameters in each path in the block (in blue) are required to be mutually-orthogonal.

---

**Algorithm 1** Example: Training a neural network with one OMP block posed on the last linear layer

---

**Require:** Training set $X_{tr} = \{(\mathbf{x}_i, y_i)\}_{i=1}^n$, number of paths $L$, orthogonality coefficient $\lambda$, learning rate $\eta$, number of epochs *epoch*, whether to perform adversarial training *adv_train*. $\theta$ denotes the to-be-optimized parameters of $h, g^1, ..., g^L$.

**Ensure:** A neural network with an OMP block: $g^i(h(\cdot)), i \in \{1, 2, ..., L\}$.

1: **while** not reach *epoch* **do**
2:     **for** $(\mathbf{x}, y)$ in $X_{tr}$ **do**
3:         $\mathcal{L}_c = \sum_{i=1}^L \mathcal{L}(g^i(h(\mathbf{x})), y); \mathcal{L}_o = \sum_{i=1}^L \sum_{j=1, j \neq i}^L \langle g^i, g^j \rangle^2$
4:         $loss = \mathcal{L}_c + \lambda \cdot \mathcal{L}_o$
5:         $\theta \leftarrow \theta - \eta \cdot \nabla_\theta loss$
6:         **if** *adv_train* **then**
7:            $\mathbf{x}^{adv} = pgd\_attack(\mathbf{x}, y, \theta)$
8:            $\mathcal{L}_{c\_adv} = \sum_{i=1}^L \mathcal{L}(g^i(h(\mathbf{x}^{adv})), y); \mathcal{L}_o = \sum_{i=1}^L \sum_{j=1, j \neq i}^L \langle g^i, g^j \rangle^2$
9:            $loss_{adv} = \mathcal{L}_{c\_adv} + \lambda \cdot \mathcal{L}_o$
10:           $\theta \leftarrow \theta - \eta \cdot \nabla_\theta loss_{adv}$
11:         **end if**
12:     **end for**
13: **end while**

---

adversarial losses are computed according to Eq.(5) to update the parameters again. More training details can be found in the appendix.

In adversarial training, when generating adversarial examples, every time the input passes through the network, one path is randomly selected from OMP block. For example, in PGD-based adversarial training, the iterative process Eq.(2) can be rewritten as

$$\hat{\mathbf{x}}^{(t+1)} = P_\gamma \Big\{ \hat{\mathbf{x}}^{(t)} + \varepsilon \mathrm{sign}(\nabla_\mathbf{x} \mathcal{L}(g^i(h(\hat{\mathbf{x}}^{(t)})), y)) \Big\}, \tag{6}$$

where the index $i$ is randomly selected from $\{1, 2, ..., L\}$.

In inference, we evaluate the performance of each single network w.r.t. every path in the OMP block, including the accuracies on clean samples and the robustness against adversarial examples.

## 4 EXPERIMENTS

### 4.1 EXPERIMENT SETUP

**Data set and network architecture.** We evaluate the performance of networks with OMP block on CIFAR10 data set (Krizhevsky, 2009), which includes 50K training samples and 10K test samples of $32 \times 32 \times 3$ color images. In both training and inference phase, the mean-variance normalization preprocess is removed for a convenient and fair comparison of different defences under $l_\infty$ bound attack on image pixel range $[0, 1]$.

The effectiveness of the OMP block is tested on two common network architectures: vgg-like network (vgg-11/13/16/19) (Simonyan & Zisserman, 2015) and residual network (resnet-20/32) (He et al., 2016). For both network architectures, we impose one OMP block on three different positions: the first convolution layer, the middle convolution layer (the middle residual block for residual network), and the last linear layer, denoted as **OMP-a**, **OMP-b** and **OMP-c** respectively. Besides, the number of paths in the OMP block is set as 10, i.e., $L = 10$.

**Adversarial attack.** For white-box attacks, we consider FGSM (Goodfellow et al., 2015) and PGD (Madry et al., 2018), two powerful and popular attacks, and impose them on every network w.r.t. each path in the OMP block. $l_\infty$ distance is used to measure the difference between the perturbed image and the clean image. For one-step FGSM, $l_\infty$ distance refers to the step size $\varepsilon$. For multi-step PGD, $l_\infty$ distance refers to the $l_\infty$ bound $\gamma$. For black-box attacks, we evaluate the robustness based on the transferability of adversarial examples. Perturbed images created from a source model are used to deceive the target model. Due to space limitation, we put the ablation study in appendix A.5.

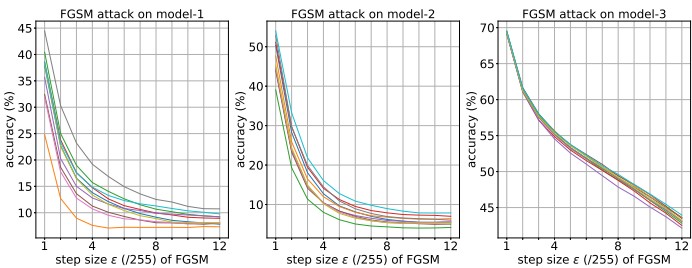

Figure 3: Robustness of each network in model-1(OMP-a), model-2(OMP-b) and model-3(OMP-c). There is a small variance among the network performance in model-3 (right), and a large variance among the network performance in model-1 and model-2 (left and middle).

## 4.2 ROBUSTNESS AGAINST WHITE-BOX ATTACK

Empirically, there is very distinct performance when OMP block is posed on the last linear layer (OMP-c) or the convolution layers (OMP-a and OMP-b). For example, suppose model-1 of OMP-a, model-2 of OMP-b and model-3 of OMP-c, then every model consists of 10 networks, each of which only differs in the paths in the OMP block. We show the results of white-box FGSM attack on these 10 networks respectively for model-1, model-2 and model-3 in Fig.3.

In Fig.3, for model-3 of OMP-c (right), the network performance shows a very small variance. Numerically, under the same step size of FGSM attack, the maximum difference among the accuracies of these 10 networks is less than two percentage points. Meanwhile, for model-1 of OMP-a and model-2 of OMP-b (left and middle), there is a much larger variance among the performance of these 10 networks. Even though the robustness of each single network in model-1 and model-2 seems much worse than model-3, but there still exist interesting properties. Next, we discuss the robustness of OMP-c separately with OMP-a and OMP-b.

**OMP block on the linear layer.** Tab.1 shows the accuracies of OMP-c on clean samples and adversarial examples. For clean samples, we report the average accuracies with standard deviations of the 10 networks of OMP-c. One can find that compared with vanilla or adversarial training network, OMP-c keeps nearly the same test accuracies on clean samples with almost no reduction. For adversarial examples, we again perform white-box FGSM and PGD attacks on each of the 10 networks in OMP-c and report the average accuracies with standard deviations. By posing one OMP block on the last linear layer, there are evident robustness improvements of OMP-c in the both cases of vanilla and adversarial training, since more robust features are learned according to Fig.1(b). Apart from attacking every single path respectively, to further verify the robustness, we also simultaneously attack all the paths in the OMP block and the results can be found in A.3.1.

**OMP block on the convolution layer.** As shown in Fig.3, there is large variance among the accuracies of networks in OMP-a and OMP-b. Besides, the robustness of single network in OMP-a or OMP-b is also vulnerable. However, suppose one network *net-1* w.r.t. one path in the OMP block gets attacked, the corresponding adversarial examples are then reclassified by another network *net-2* w.r.t. another path in the OMP block. Under this setting, accuracies of *net-2* on the adversarial examples get obviously improved compared with *net-1*, which indicates that only by changing the specified layer with another orthogonal parameters, network robustness can get enhanced.

For OMP-a, where the OMP block is posed on the first layer, we select one path and perform FGSM and PGD attacks on this network. The corresponding accuracy variations are shown as the dashed line in Fig.4(a). The adversarial examples created from this network are then reclassified by other networks w.r.t. other paths in the OMP block, which shows significantly improvements as the solid lines in Fig.4(a). There also exists similar phenomenon of OMP-b, shown in Fig.4(b).

We give a brief summary for the properties of OMP-a, OMP-b and OMP-c:

- For OMP-c, which consists of mutually-orthogonal linear classifiers, there is a small performance variance among these networks. The robustness of each network all gets sig-

Table 1: Accuracies (%) on clean samples and adversarial examples of test set in CIFAR10. The $l_\infty$ distance of FGSM and PGD attacks equals to $8/255$.

| model | vanilla training | | adv. training | | |
|---|---|---|---|---|---|
| | clean | FGSM | clean | FGSM | PGD |
| vgg11 | 91.33 | 19.59 | 87.04 | 41.52 | 25.81 |
| OMP-c | 91.57±0.02 | **30.27±0.21** | 86.73±0.03 | **43.95±0.04** | **27.53±0.08** |
| vgg13 | 93.19 | 25.93 | 89.23 | 48.67 | 32.48 |
| OMP-c | 93.17±0.02 | **35.12±0.17** | 89.28±0.03 | **50.60±0.03** | **33.86±0.06** |
| vgg16 | 93.18 | 19.52 | 88.77 | 56.51 | 44.23 |
| OMP-c | 93.02±0.01 | **49.04±0.47** | 89.01±0.01 | **62.42±0.03** | **50.60±0.10** |
| vgg19 | 92.86 | 10.35 | 89.07 | 55.25 | 39.30 |
| OMP-c | 92.83±0.01 | **53.00±0.71** | 88.95±0.01 | **63.35±0.05** | **54.52±0.10** |
| resnet20 | 91.71 | 13.36 | 88.03 | 35.06 | 22.26 |
| OMP-c | 91.75±0.03 | **25.58±0.66** | 87.95±0.03 | **38.24±0.10** | **23.64±0.08** |
| resnet32 | 93.20 | 28.26 | 88.65 | 37.93 | 23.26 |
| OMP-c | 92.85±0.03 | 24.15±0.59 | 89.14±0.04 | **43.12±0.16** | **26.91±0.14** |

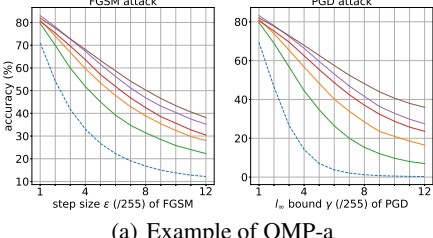

(a) Example of OMP-a

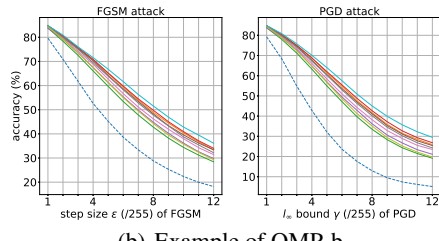

(b) Example of OMP-b

Figure 4: Illustration of the robustness of OMP-a and OMP-b. Single network in OMP-a or OMP-b shows terrible robustness (dashed lines). But adversarial examples created from this network could be classified correctly by other networks (solid lines). Detailed settings can be found in A.3.

nificantly improved, with no performance reduction on clean samples, which indicates the possibility of balancing the trade-off between generalization and adversarial robustness.

- For OMP-a and OMP-b, which consist of mutually-orthogonal convolution layers, there is a large performance variance among these networks. The robustness of single network is terrible. But, adversarial examples created from one of these networks can be successfully reclassified by other networks. It indicates that by altering the direction of the parameters of one layer to another orthogonal direction, network robustness can get improved. To the best of our knowledge, this is also the first work to enhance the network robustness by only changing the parameters of one layer.

### 4.3 ROBUSTNESS AGAINST BLACK-BOX ATTACK

We evaluate the robustness of network with OMP block against black-box attacks based on the transferability of adversarial examples. In detail, we perform FGSM and PGD attacks on two transparent source models: vanilla vgg16 and resnet20. The resulting adversarial examples are then reclassified by the target model. Target models include vanilla and adversarial training networks and the corresponding OMP-c. Results against black-box attacks from different attacks and network architectures are shown in Tab.2. Based on the improved or competitive accuracies in Tab.2, OMP-c could successfully defend black-box attacks. Results of networks with OMP-a and OMP-b can be found in the appendix A.4.

Table 2: Accuracies (%) of black-box attacks on test set in CIFAR10. The $l_\infty$ distance of FGSM and PGD attacks equals to $8/255$.

| target model | attacks on source models | | | |
| | FGSM on vgg16 | PGD on vgg16 | FGSM on resnet20 | PGD on resnet20 |
|---|---|---|---|---|
| vgg16 | 19.52 | 0.02 | 49.57 | 46.81 |
| OMP-c | **41.91±0.04** | **25.46±0.03** | 49.44±0.03 | **49.55±0.02** |
| vgg16-adv | 85.13 | 86.92 | 86.67 | 87.46 |
| OMP-c-adv | **85.37±0.01** | **87.28±0.01** | **86.78±0.01** | **87.84±0.01** |
| resnet20 | 37.64 | 19.36 | 13.36 | 0.00 |
| OMP-c | 36.79±0.06 | **21.06±0.04** | **32.66±0.09** | **18.08±0.05** |
| resnet20-adv | 82.98 | 85.34 | 84.50 | 86.22 |
| OMP-c-adv | 82.43±0.05 | 85.18±0.03 | 84.21±0.04 | 86.19±0.07 |

## 4.4 ROBUSTNESS COMPARISON

The robustness of networks with proposed OMP block is compared with two defence methods via imposing randomness on parameters, PNI (He et al., 2019) and advBNN (Liu et al., 2018b). Both methods aim to learn a distribution for the parameters. PNI adds learnable Gaussian noise on parameters. AdvBNN incorporates adversarial training into Bayesian neural network.

Accuracies of different models against white-box FGSM and PGD attacks are shown in Fig.5. These models are of vgg-like architecture. We compare networks with OMP block pose on the last linear layer with other methods, and report the average accuracies and standard deviations of OMP-c. It is clearly seen that in vgg-like model, networks imposed orthogonality show much better robustness compared with networks imposed randomness on parameters.

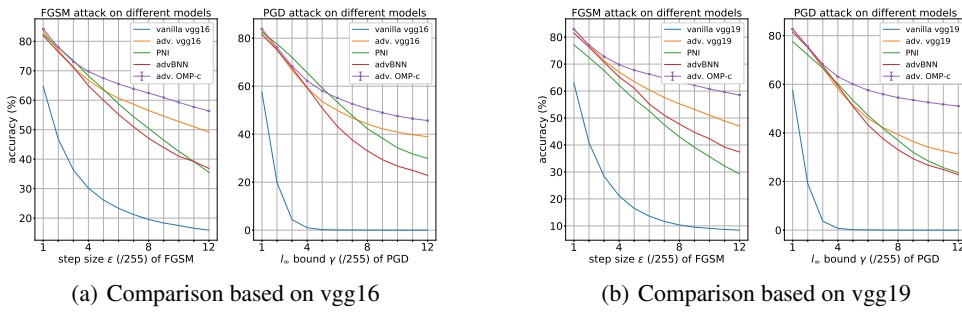

(a) Comparison based on vgg16          (b) Comparison based on vgg19

Figure 5: Robustness comparison of different models against FGSM and PGD attacks.

## 5 CONCLUSION

In this work, we propose to embed orthogonal multiple paths into a neural network to enhance robustness. The mechanism behind is to learn robust features via forcing the follow-up layers to learn to fit all the paths and correcting the features learned by front layers. We investigate distinct properties of different positions of the OMP block against white-box attacks, e.g., balancing the trade-off between generalization and robustness, and enhancing robustness via only changing one-layer parameters. Besides, experiments on defending black-box attacks, robustness comparison and ablation study are all adopted to verify the effectivenss of OMP block. We hope that this work could help inspire researchers to better understand the essence of generalization and robustness of DNNs.

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

## A  APPENDIX

### A.1  TRAINING

We first give more details in training networks with OMP block:

- When computing Eq.(5), we determine an equally-weighted loss with respect to each path in the OMP block, i.e., the loss term in Eq.5 actually is $\sum_{k=1}^{L} \frac{1}{L} \mathcal{L}(g^k(h(\mathbf{x})), y)$.
- For the orthogonality coefficient $\lambda$ in Eq.5, we set this hyper-parameter as $\lambda = 0.1$.

We do not investigate more alternative values since this setting already results in satisfying performance.

## A.2 NETWORK STRUCTURE

Then, we describe the detailed structures of networks with OMP block:

- For network of OMP-a, the OMP block is posed on the first convolution layer. In vgg-like network, each path in OMP block contains the convolution kernel parameters of $3 \times 64 \times 3 \times 3$ dimension. In residual network, each path in OMP block contains the convolution kernel parameters of $3 \times 16 \times 3 \times 3$ dimension. The bias term in the first convolution layer is omitted.

- For network of OMP-b, the OMP block is posed on the middle convolution layer. In vgg-like network, the OMP block is posed on the convolution layer right after the first max-pooling layer, and each path contains the kernel parameters of $64 \times 128 \times 3 \times 3$ dimension. In residual network, the OMP block is posed on the second residual block in the second stage, and each path contains a residual block. Only the convolution parameters in the residual block are required to be mutually-orthogonal. The bias term in the convolution layer is omitted.

- For network of OMP-c, the OMP block is posed on the last linear layer. In vgg-like network, each path in OMP block contains the weight matrix of $512 \times 10$ dimension. In residual network, each path in OMP block contains the weight matrix of $64 \times 10$ dimension. The bias term in the last linear layer is omitted.

## A.3 ROBUSTNESS AGAINST WHITE-BOX ATTACK

We explain how we produce the results in Fig.4. Results in Fig.4(a) and Fig.4(b) are based on resnet20 architecture. We first show the accuracies on training and test sets of every resnet20 network w.r.t. each path in OMP-a or OMP-b block respectively in Tab.3.

Table 3: Accuracies (%) of each network w.r.t. each path in OMP block on CIFAR10 clean images.

|  |  | net-1 | net-2 | net-3 | net-4 | net-5 | net-6 | net-7 | net-8 | net-9 | net-10 |
|---|---|---|---|---|---|---|---|---|---|---|---|
| OMP-a | training | 95.26 | 96.108 | 93.022 | 13.914 | 68.96 | 97.42 | 96.464 | 85.98 | 84.576 | 94.104 |
|  | test | 87.11 | 87.62 | 85.24 | 13.82 | 66.05 | 88.86 | 87.32 | 79.72 | 78.68 | 85.59 |
| OMP-b | training | 98.868 | 98.81 | 98.88 | 98.948 | 98.894 | 98.798 | 98.912 | 98.918 | 98.814 | 98.906 |
|  | test | 88.28 | 88.14 | 88.01 | 88.44 | 88.15 | 88.35 | 88.16 | 88.2 | 88.28 | 88.28 |

In Tab.3, for OMP-a, some networks show distinct accuracies on clean samples, e.g. net-4, net-5, net-8 and net-7. While every network in OMP-b shows similar accuracies on clean samples. However, the robustness of these networks is vulnerable and varies a lot (see the left and middle images in Fig.3).

In Fig.4(a), the attacked network of OMP-a (dashed lines) is the net-6 in the table above, which shows the highest accuracies on clean samples. We record the accuracies of net-1, net-2, net-3, net-7 and net10 (solid lines) on the adversarial examples created from net-6. Other networks are not considered since their performance on clean samples is not acceptable.

In Fig.4(b), the attacked network of OMP-b (dashed lines) is also the net-6 in the table above. We record the accuracies of the other 9 networks (solid lines) on the adversarial examples created from net-6.

### A.3.1 A STRONGER WHITE-BOX ATTACK

In this section, we perform a stronger white-box attack to further verify the robustness of network with OMP-c. Instead of attack every single network w.r.t. every path in the OMP block, we simultaneously attack all the networks w.r.t. all paths in the OMP block. The resulting adversarial examples are then reclassified by all the networks. The average accuracies and standard deviations against FGSM and PGD attacks are shown in Fig.6. It can be seen that even though all the paths are attacked simultaneously, networks with OMP block could still defend this stronger white-box attack and sustain better robustness than adversarial training networks.

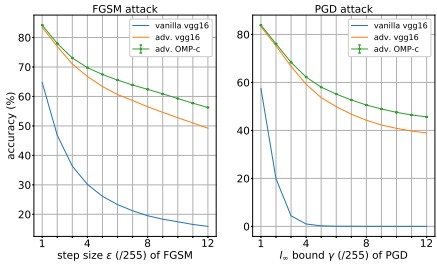 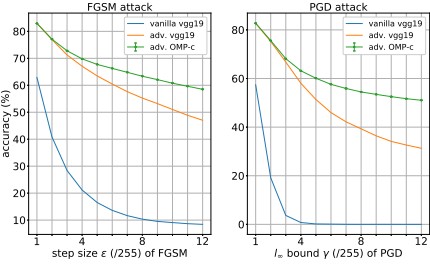

(a) Robustness of vgg16 architecture        (b) Robustness of vgg19 architecture

Figure 6: Robustness against a stronger white-box attack, which simultaneously attacks all the paths in the OMP block. Even under such a stronger attack, the networks of OMP-c block still show superior robustness.

## A.4 ROBUSTNESS AGAINST BLACK-BOX ATTACK

We add the results of networks with OMP-a and OMP-b block against black-box attacks in Tab.4.

In Tab.4, for networks with OMP-a block, since each network w.r.t. each path shows a large variance, we only report the largest accuracies among these networks. For networks with OMP-b block, we still report the average accuracies of all networks w.r.t. all paths in the block together with the standard deviations. These results indicate that networks with OMP-a or OMP-b both have the ability in defending black-box attacks.

Table 4: Accuracies (%) of black-box attacks on test set in CIFAR10.

| target model | attacks on source models | | | |
| --- | --- | --- | --- | --- |
| | FGSM on vgg16 | PGD on vgg16 | FGSM on resnet20 | PGD on resnet20 |
| vgg16 | 19.52 | 0.02 | 49.57 | 46.81 |
| OMP-a | 30.92 | **27.39** | 30.26 | 38.31 |
| OMP-b | 35.88±2.10 | 25.84±1.20 | 37.02±2.45 | 41.78±2.00 |
| OMP-c | **41.91±0.04** | 25.46±0.03 | 49.44±0.03 | **49.55±0.02** |
| vgg16-adv | 85.13 | 86.92 | 86.67 | 87.46 |
| OMP-a-adv | 61.41 | 64.57 | 64.88 | 68.42 |
| OMP-b-adv | 80.34±1.39 | 83.41±1.46 | 82.72±1.53 | 84.88±1.44 |
| OMP-c-adv | **85.37±0.01** | **87.28±0.01** | **86.78±0.01** | **87.84±0.01** |
| resnet20 | 37.64 | 19.36 | 13.36 | 0.00 |
| OMP-a | **39.35** | **36.02** | **38.23** | **39.67** |
| OMP-b | 34.81±0.25 | 18.96±0.10 | 31.88±0.23 | 18.18±0.16 |
| OMP-c | 36.79±0.06 | 21.06±0.04 | 32.66±0.09 | 18.08±0.05 |
| resnet20-adv | 82.98 | 85.34 | 84.50 | 86.22 |
| OMP-a-adv | 77.71 | 83.40 | 80.71 | 85.24 |
| OMP-b-adv | 82.70±0.20 | **85.46±0.14** | 84.32±0.19 | **86.41±0.18** |
| OMP-c-adv | 82.43±0.05 | 85.18±0.03 | 84.21±0.04 | 86.19±0.07 |

## A.5 ABLATION STUDY

In this subsection, we explain the necessity of learning the OMP-block. Empirically, once the coefficient $\lambda = 0$, then the parameters of multiple paths in OMP block will converge to the same one, which is meaningless. Therefore, we adopt the ablation study from another perspective. Actually, two random vectors in high-dimensional space are apt to be orthogonal, in other words, the inner product of two random high-dimensional vectors approximates zero. Hence we fix the parameters

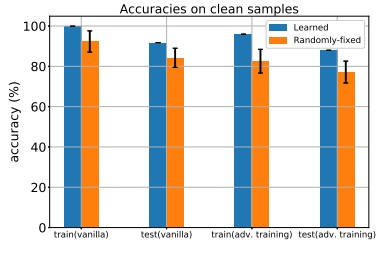 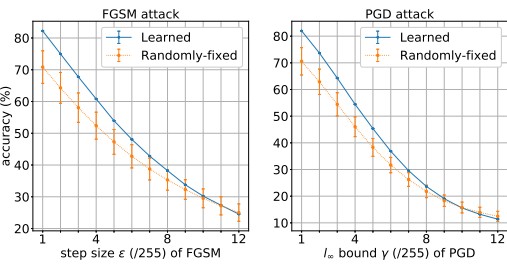

(a) Performance on clean samples

(b) Robustness against adversarial attack

Figure 7: Comparison between networks with learned OMP block (in blue) and networks with randomly-fixed OMP block (in orange). The network architecture is resnet20 of OMP-c. The average mean accuracies of all the networks and the standard deviations are recorded.

of the OMP block with random values and only train the rest of the neural network. In this way, we could verify the effectiveness of backward correction.

Fig.7 shows the comparison results between networks with learned OMP block and networks with randomly-fixed OMP block. Fig.7(a) illustrates the performance on clean training and test samples. Fig.7(b) illustrates the robustness against FGSM and PGD attacks. If the OMP block is not jointly learned, there is a large variance among the network performance. Besides, the performance of network with randomly-fixed OMP block on both clean samples and adversarial examples also becomes worse than that of network with learned OMP block. Therefore, only by jointly learning the embedded OMP block with the remainder of the network, the network could get robustness improvements and keep high accuracy on clean samples simultaneously.

