# OpenReview forum: "Learn Robust Features via Orthogonal Multi-Path"
_ICLR.cc/2021/Conference — Reject_

### Official Review · AnonReviewer3 · 2020-10-28
**Experiments are somewhat limited**

**Rating:** 5
**Confidence:** 3

**Review:**

This paper addresses the problem of adversarial defence, by proposing to build multiple parallel orthogonal layers to replace a regular neural network layer. The layers in the OMP block are trained to be diverse and orthogonal to each other. Experiments on both white-box and black-box attacks, with or without adversarial training, have been carried out to show the efficacy of the proposed method, on the CIFAR 10 dataset.

\- I think for OMP-a and OMP-b, it is no longer considered as white-box attack when we change the direction of the parameters of one layer to another orthogonal direction, as the authors described below Fig 4.

\- Only Cifar 10 is used to verified the proposed method. Since Cifar 10 is a simpler dataset with low resolutions, it might be helpful to involve other datasets in the evaluation part.

\- There are some other defenses based on diversity ensemble, which are not discussed in the paper. For example, "Improving Adversarial Robustness via Promoting Ensemble Diversity".

\- From Table 2, does it mean the proposed method is not effective to improve the robustness for black-box attack compared with vanilla adversarial training?

After reading the response, my concerns are not fully resolved. For example, based on "For OMP-a and OMP-b, ... The robustness of single network is terrible. ... adversarial examples created from one of these networks can be successfully reclassified by other networks", I am feeling the OMP-a and OMP-b are less effective based on a real white-box attack. Also some other concerns are not fully addressed. Thus I am keeping my rating unchanged.

---

> ### Author Response · Authors · 2020-11-22
> **Response to Review3**
>
> Thank you for the constructive review. Concerning your remarks:
>
> ## 1.“it is no longer considered as white-box attack”.
>
> OMP-a/b/c only are only different to the place the OMP block is imposed. After training, only one path in the OMP is chose and then the interference is same as the regular networks. Then white-box attacks are imposed to the obtained network. (It is not the case that we first do white-box attack on the vanilla network and then change the layer.)
>
> ## 2.“Only CIFAR10”.
>
> According to the comments, we have tested our method on **CIFAR100** data set.
>
> We give the results of vgg16/19 trained on **CIFAR100**.
>
> The step size of FGSM and bound of PGD are both $8/255$.
>
> For CIFAR100, we set the number of paths in OMPc as 40.
>
> - vanilla training
>
> model | clean | FGSM
> :-:         | :-:       | :-:
> vgg16  | 72.45 | 16.09
> OMP-c | **73.05** | **22.24**
> ----------|---------|-------------
> vgg19  | 69.90 | 16.07
> OMP-c | **71.76** | **30.13**
>
> - adversarial training
>
> model | clean | FGSM | PGD
> :-:         | :-:       | :-:        | :-:
> vgg16  | 62.62 | 22.80  | 13.90
> OMP-c | **62.87** | **25.86** | **15.90**
> ----------|---------|---------|---------
> vgg19  | 60.90 | 19.93 | 11.25
> OMP-c | **62.08** | **33.13** | **24.27**
>
> For models trained on CIFAR100, OMP-c could still enhance robustness.
>
> ## 3.“defenses based on diversity ensemble”.
>
> As answered to your 1st question, only one path is chosen after training and hence OMP is not an ensemble strategy. Actually, for ensemble methods, the white-box attack is not convenient to impose (simply imposing is not white-box attack) and hence will be unfair (also considering about the model size and inference time).
>
> ## 4.“the results on Table 2”.
>
> In Table 2, when the source model and the target model are of the same structure, OMP model is much more effective than vanilla training model in defending black-box attacks. Meanwhile, in other cases, such as where adversarial training is included, and where the source model and the target model are not of the same structure, OMP model indeed only shows slightly better or similar performance compared with vanilla adversarial training model and is expected to be further improved.

---

### Official Review · AnonReviewer1 · 2020-10-29

**Rating:** 5
**Confidence:** 3

**Review:**

The paper proposes an adversarial example defense method based on ensembling the final linear layer of a classifier, where the components of the ensemble are jointly trained with each other and with the backbone of the model to minimize cross-entropy and to be approximately orthogonal with each other. Orthogonality here is measured by flattening the layers weight matrices and computing their inner products. Training examples are either original classification examples or adversarial examples computed by the PGD attack, used in a generative adversarial traning approach.

The authors experiment on standard image classification dataset and compute robustness to both white box and black box attacks, obtaining some improvements over plain GAT. The authors also experiment with ensembling other layers of the model but obtain worse results.

The proposed method is interesting, however it increases the model size, hence it should be compared to a non-robust or GAT model for the same parameter budget. Also, in the description of the inference procedure it was not very clear how the ensemble predictions are combined. Overall, this appears to be an incremental improvement.

---

> ### Author Response · Authors · 2020-11-22
> **Response to Review1**
>
> Thank you for the constructive review. Concerning your remarks:
>
> ## 1.“increasing model size”.
> The proposed OMP is NOT an ensemble strategy. OMP block is added in the training procedure and only one path is selected in the inference. The robustness comes from the adaptiveness of the networks not from ensemble. Therefore, the model size is as the SAME as the original networks.
>
> Indeed, in the training phase, we need to train multiple paths. But in the experiments, we add OMP only in one layer and hence the burden is increased very little. For example, the parameters in original VGG16 is 14.73M. When an OMP block containing 10 paths is posed on the last linear layer, the parameter size is 14.77M in the training phase and still 14.73M in the inference phase.
>
> ## 2.“description of the inference procedure”.
>
> In our paper, we do NOT perform inference via any ensemble strategy. After training the OMP block, only one path is chosen in the inference procedure. That is the inference procedure is as the same as the original network. Maybe the misunderstanding is due to our description and we will try to make the description clearer in the revised version.

---

### Official Review · AnonReviewer2 · 2020-10-30
**A different adversarial training approach**

**Rating:** 5
**Confidence:** 3

**Review:**

This paper proposes to learn multiple near-orthogonal paths (OMP) in the CNN which could provide better adversarial training performance by using one random path selected from the OMP block, improving the diversity of the adversarial training examples generated. Results show some improvements over regular adversarial training. Interestingly, the improvements are very significant on the VGG networks, while not quite significant for the ResNet variants tested.

This paper claimed that it creates orthogonal paths, but it's realistically near-orthogonal since they only added a soft constraint on the OMP regularization term, similar algorithms have been proposed in the past:

[Bansal et al. 2018] Can we gain more from orthogonality regularizations in training deep cnns?

There have also been quite a few work on learning real orthogonal paths based on Riemannian manifold optimization. Some of these are of similar speed as conventional SGD and Adam. A review paper can be found at:

[Huang et al. 2020] Normalization Techniques in Training DNNs: Methodology, Analysis and Application.

Some of those papers should be cited.

In terms of performance, I feel this work should be compared against other regularization-based adversarial defense methods. A couple examples of that are:

Qin et al. Adversarial Robustness through Local Linearization. NeuRIPS 2019
Mao et al. Metric Learning for Adversarial Robustness. NeuRIPS 2019.

Comparisons against those algorithms would further verify the performance of the proposed approach.

Besides, there should be some discussions on potentially why the improvements on VGG networks are very significant and not so much on ResNet.

There is also some recent evidence on the effect of early stopping on adversarial defenses (e.g. Rice et al. Overfitting in adversarially robust deep learning. ICML 2020). It would be nice if the authors could state when did they stop the training of the respective models.

In terms of ablation, it would be nice to see different inference schemes. e.g. whether using a subset of the paths in the OMP block would be beneficial against adversarial examples or not.

I look forward to seeing the authors rebuttal and comments from other reviewers.

---

> ### Author Response · Authors · 2020-11-22
> **Response to Review2**
>
> Thank you for the constructive review. Concerning your remarks:
> ## 1."Some of those papers should be cited"
> Thanks for the provided relevant papers [1,2]. These papers provide some theoretical insights about the orthogonality constraint. We will add citations to these papers in the revised version. The proposed OMP model is to enhance the adaptiveness of the neural networks to different paths. To guarantee the differences, we require the parameters of the paths are orthogonal by adding a, we admit, very simple constraint. We agree with you that they are only near-orthogonal. But it already achieves our aim. But of course, we expect that with other ingenious design on the orthogonality, the performance could be further improved in the future study.
>
> ## 2.“Compared with regularization-based defense”.
>
> The orthogonal constraint is not added in the training target as a regularization term. Instead, it is used to guarantee the difference of the paths, and then to correct the features learned by front layers, resulting in more robust features.
>
> Therefore, the proposed OMP model is not a typically regularization-based defense. Generally, regularization-based defenses have accuracy loss in clean examples but the proposed OMP could keep the performance. Indeed, from the view of model capacity controlling, OMP could be viewed as regularization and so are the compared algorithms, namely RSE [3], PNI [4] and advBNN [5].
>
> ## 3.“discussions on different improvements between VGG and ResNet”.
>
> Indeed, this is an interesting point and deserves discussion. VGG consists of several cascaded convolutional layers, while ResNet owns the unique residual connection structure. Perhaps residual connection structure is helpful for learning more robust features, which results in not-so-significant improvements on Resnet. However, this guess is quite heuristic and thus we did not include in the manuscript.
>
> ## 4. “early stopping”.
> Thanks for providing the reference about early stopping. In our OMP training, we do not use early stopping trick, and just stop training the network after a fixed number of epochs. In detail, for vgg, we train the networks on the training set for 200 epochs. For Resnet, the number of training epochs is 350. We will add this setting in the appendix and will discuss early stopping in the future.
>
> ## 5.“different inference schemes”.
>
> This is an interesting suggestion. Currently we do not use any ensemble strategy and report the performance of each individual network w.r.t. each path. We will test the performance of more inference schemes later.
>
> ##
>
> [1] Bansal et al. 2018. Can we gain more from orthogonality regularizations in training deep cnns?;
>
> [2] Huang et al. 2020. Normalization Techniques in Training DNNs: Methodology, Analysis and Application
>
> [3] Liu et al. Towards robust neural networks via random self-ensemble. ECCV 2018
>
> [4] He et al. Parametric noise injection: Trainable randomness to improve deep neural network robustness against adversarial attack. CVPR 2019
>
> [5] Liu et al. Adv-bnn: Improved adversarial defense through robust Bayesian neural network. ICLR 2018

---

### Official Review · AnonReviewer4 · 2020-10-31
**Official Blind Review #4**

**Rating:** 4
**Confidence:** 5

**Review:**

In this work, the authors propose to use orthogonal multi-path(OMP) block to improve the adversarial robustness of deep neural networks. They introduce three types of OMP, OMP-a/b/c, based on where the OMP block is located in the neural network. Experimental results demonstrate the effectiveness of their OMP approach. The idea is interesting and the paper is easy to follow.

However, I have some concerns below:

1.	I feel the contribution of this work is not sufficient as the orthogonal feature learning has been explored in natural training.
2.	In addition, the baselines are also not enough for the convincing. From the perspective of network structure, this work needs to compare with related work of network structure in adversarial robustness studies, such as Feature Denoising [1]. From the perspective of model ensembling or diverse feature learning (since OMP is an ensemble strategy), this work needs to compare with works on model ensembling or diverse feature learning [2]
3.	The comparison is rather limited. The experiments are only conducted on one dataset, CIFAR-10, and the attacks for evaluation are limited.

[1] Feature Denoising for Improving Adversarial Robustness. CVPR 2019.
[2] Improving Adversarial Robustness via Promoting Ensemble Diversity. ICML 2019.

---

> ### Author Response · Authors · 2020-11-22
> **Response to Review4**
>
> Thank you so much for taking the time to review the paper and we appreciate the comments! Concerning your remarks:
> ## 1.“not sufficient contribution as the orthogonal feature learning has been explored”.
> Our work is different from orthogonal feature learning, which is to learn a group of orthogonal features and then ensemble them. Instead, we are going to train features that are adaptive to different situations, i.e., a group of orthogonal parameters (OMP block). In orthogonal feature learning, the robustness comes from ensemble strategy and then the orthogonality is to ensure the diversity. In the proposed method, the robustness comes from the adaptiveness of the networks, since it is trained considering together orthogonal paths.
> ## 2.“baselines are not enough for the convincing.”
> Thanks for the suggestions. Accordingly, we have compared OMP with Feature Denoising [1] (Feature Denoising for Improving Adversarial Robustness) and the improvement is quite significant (below are the average accuracy with 10 trials.)
>
> Due to time limitation, results of other structure and on other datasets will be reported in the final version. Besides, as answered above, our OMP is NOT an ensemble strategy. For defense performance evaluation under white-box attack, it is somehow not fair and not convenient to compare with ensemble methods (for ensembled classifier, the white-box attack should be modified.)
>
> Robustness against FGSM attack [vgg19]
>
> FGSM-$\epsilon$ | 0 | 1/255 | 2/255 | 3/255 | 4/255 | 5/255 | 6/255 | 7/255 | 8/255 | 9/255 | 10/255 | 11/255 | 12/255
> :-: | :-:            | :-:   | :-:   | :-:   | :-:   | :-:   | :-:   | :-:   | :-:   | :-:   | :-:    | :-:    | :-:
> [1]    | 89.17|83.81|77.42|70.55|63.49| 56.35 | 50.15 | 44.59 | 39.47 | 35.69 | 31.88 | 28.49 | 25.37
> OMP-c |88.95|82.93|77.08|**72.79**|**69.79**| **67.74** | **66.27** | **64.83** | **63.35** |**62.12** | **60.79** | **59.63** | **58.48**
>
> Robustness against PGD attack [vgg19]
>
> PGD-$\delta$    | 0 | 1/255 | 2/255 | 3/255 | 4/255 | 5/255 | 6/255 | 7/255 | 8/255 | 9/255 | 10/255 | 11/255 | 12/255
>  :-: | :-:            | :-:   | :-:   | :-:   | :-:   | :-:   | :-:   | :-:   | :-:   | :-:   | :-:    | :-:    | :-:
> [1]	   |89.17|83.65|76.47|67.50|57.48 | 47.73 | 39.08 | 32.26 | 26.23 | 21.65 | 17.88 | 15.48 | 13.43
> OMP-c |88.95|82.71|75.60|**68.20**| **63.16** | **60.07** | **57.54** | **55.90** | **54.52** | **53.51** | **52.56** | **51.72** | **51.05**
>
> ##  3.“only conducted on one dataset, CIFAR-10, and the attacks for evaluation are limited.”
>
> Thanks for the comments. We have evaluated the method against another common l2-norm based attack, **DeepFool** attack, and on **CIFAR100**. The results are reported here.
>
> ### 3.1.  results against **DeepFool** attack with 3 steps of vgg13/19 trained on CIFAR10.
>
> Model | vanilla training | adv. Training |
> :-:         | :-:                | :-:                 |
> vgg13 | 5.48		   | 9.94		   |
> OMP-c| **7.57**		   | **11.32**	   |
> ---------| ------------------| -----------------|
> vgg19 | 5.15		   | 9.94		   |
> OMP-c| **5.50**		   | **10.19**	   |
>
> ### 3.2. results of vgg16/vgg19 trained on **CIFAR100**
>
> The step size of FGSM and bound of PGD are both $8/255$. For CIFAR100, we set the number of paths in OMP-c as 40.
>
>  - vanilla training
>
> Model | clean    |  FGSM 	   |
> :-:         | :-:          | :-:               |
> vgg16  | 72.45    | 16.09         |
> OMP-c | **73.05**    | **22.24**	   |
> -------- | -----------|---------------|
> vgg19 | 69.90    | 16.07	   |
> OMP-c | **71.76**    | **30.13**	   |
>
>   - adversarial training
>
> Model | clean    |  FGSM 	   |  PGD     |
> :-:         | :-:          | :-:               | :-:          |
> vgg16  | 62.62    | 22.80         |13.90
> OMP-c | **62.87**    | **25.86**	  |**15.90**
> -------- | -----------|---------------| ---------
> vgg19 | 60.90    | 19.93	  | 11.25
> OMP-c | **62.08**    | **33.13** 	  | **24.27**
>
> ##
> [1] Feature Denoising for Improving Adversarial Robustness. CVPR2019

---

### Decision · Program_Chairs · 2021-01-07
**Final Decision**

**Decision:**

Reject

**Comment:**

I thank the authors and reviewers for the lively discussions. Although reviewers mentioned the work has potentials to improve adversarial robustness, they agreed that the current draft needs a bit more work specially to strengthen its experimental results and comparisons with related works.